# A Systematic Review Protocol to Identify the Key Benefits and Efficacy of Nature-Based Learning in Outdoor Educational Settings

**DOI:** 10.3390/ijerph18031199

**Published:** 2021-01-29

**Authors:** Jeff Mann, Tonia Gray, Son Truong, Pasi Sahlberg, Peter Bentsen, Rowena Passy, Susanna Ho, Kumara Ward, Rachel Cowper

**Affiliations:** 1Centre for Educational Research, Western Sydney University, Sydney, NSW 2751, Australia; t.gray@westernsydney.edu.au; 2The Scots College, Bellevue Hill, Sydney, NSW 2023, Australia; 3Faculty of Health, School of Health and Human Performance, Dalhousie University, Halifax, NS B3H 4R2, Canada; son.truong@dal.ca; 4Gonski Institute, University of New South Wales, Sydney, NSW 2052, Australia; Pasi.Sahlberg@unsw.edu.au; 5Center for Clinical Research and Prevention, Copenhagen University Hospital, Bispebjerg and Frederiksberg, 2000 Frederiksberg, Denmark; peter.bentsen@regionh.dk; 6Department of Geosciences and Natural Resource Management, University of Copenhagen, 1958 Frederiksberg C, Denmark; 7Plymouth Institute of Education, University of Plymouth, Drake Circus, Plymouth PL4 8AA, UK; R.Passy@plymouth.ac.uk; 8Singapore University of Social Sciences, Singapore 599494, Singapore; susannaho001@suss.edu.sg; 9School of Education and Social Work, University of Dundee, Nethergate, Dundee DD1 4HN, UK; kward001@dundee.ac.uk; 10Thrive Outdoors, Edinburgh EH11 3AF, UK; rachel@inspiringscotland.org.uk

**Keywords:** Outdoor Learning, health, natural environments, socio-emotional, systematic review, protocol, wellbeing

## Abstract

Outdoor Learning in natural environments is a burgeoning approach in the educational sector. However, the evidence-base of research has not kept pace with teacher perceptions and increased practitioner usage. Anecdotal evidence and formal research suggest the significant health and wellbeing benefits of nature connection. Offering low-cost, non-invasive pedagogical solutions to public health challenges—particularly around mental health, wellbeing, physical literacy, and increasing physical activity–the pedagogical benefits of Outdoor Learning are yet to be fully enunciated. The proposed systematic review will search for studies across eight academic databases which measure the academic and socio-emotional benefits of Outdoor Learning, with a focus on school-aged educational settings. Using the inclusion criteria set out in this paper (and registered with PROSPERO: CRD42020153171), relevant studies will be identified then summarised to provide a synthesis of the current literature on Outdoor Learning. The goal of this review is to document the widespread international investigation into Outdoor Learning and its associated benefits for development, wellbeing, and personal growth. The systematic review will provide insights for teacher-training institutions, educational policy makers, and frontline teachers to improve the learning experiences of future students.

## 1. Introduction

Learning outdoors has been the norm through most of human history. Gaining an understanding of how to thrive and maintain a balance in the natural environment was the learning context for Indigenous cultures across the world [1,2]. When mass education began in the 19th century, the setting for learning moved indoors to the classroom even though playing outdoors was commonplace [3]. Many still saw the need for learning to be connected in the natural world, however, and by the latter half of the 20th century ‘outdoor education’ sprang up in many countries as an alternative non-curriculum based form of learning [4,5,6,7,8,9,10]. Unfortunately outdoor education has been largely marginalized from mainstream curriculum-based schooling [11,12]. Outdoor education research has demonstrated benefits of nature-based adventure experiences for intrapersonal and interpersonal growth [13,14,15], but we are curious about whether academic learning in outdoor environments could also have a positive impact for curriculum-based education?

### 1.1. Conceptualisation of Outdoor Learning

Learning outdoors has been defined as taking students outdoors into their immediate or nearby surroundings to learn essential lessons of the curriculum, with four possible zones: (a) school grounds, (b) local neighbourhoods, (c) day excursions, and (d) overnight stays/residential camps and expeditions [16]. Teachers incorporate the local natural environment to teach specific school subjects [17], and students are encouraged to make their own choices and decisions, develop mastery, and build a sense of purpose and meaning [18]. We specifically define Outdoor Learning as regular and structured learning experiences for school-aged children in on-campus or off-campus outdoor settings. Out-of-school learning [19], by contrast, incorporates school field trips to a variety of indoor and outdoor off-campus learning environments including museums and industrial plants.

### 1.2. Outdoor Learning: Key Literature

The literature on Outdoor Learning, as we have defined above, is generally sparse, however four examples stand out. Firstly, Scottish outdoor educators Beames, Higgins, and Nicol [16] wrote a seminal book titled *Learning Outside the Classroom*, with the aim of helping mainstream teachers “incorporate more meaningful Outdoor Learning opportunities into their daily teaching activities” (inside cover). This manual provides classroom teachers with an overview of educational theory supporting Outdoor Learning, and a practical guide for how to use school grounds and local neighbourhoods as an extension of their classroom [20].

Elsewhere in the UK, the Natural Connections Demonstration Project [21] used a distributed model of brokerage to recruit 125 schools in southwest England to develop Outdoor Learning programs. School surveys reflected that students found Outdoor Learning enjoyable and engaging, and that it had positive impacts on their connection with nature, social skills, health and wellbeing, and academic attainment. Although teachers’ lack of confidence to teach outside was initially a challenge, teachers reported positive impacts on their professional practice, health and wellbeing, and job satisfaction.

The third landmark for Outdoor Learning is Denmark’s concept of ‘udeskole’ (outdoor school), where students spend regular hours every week learning outside (irrespective of the weather). Denmark is often recognised as a world leader in Outdoor Learning [18], and its recent national TEACHOUT pilot has provided research evidence of Outdoor Learning benefits to primary school students’: physical activity [22], motivation for school [23], prosocial behaviour [24], and even reading performance [25]. The Forest School movement in the UK was initially inspired by the Danish practice of Outdoor Learning [26], and emphasises child-centred learning for early learning and primary school aged children in a forest environment [27].

In terms of a comprehensive overview of Outdoor Learning research around the world, the final highlight is a systematic review of ‘regular classes in outdoor settings’ [28]. This paper potentially makes the current review redundant, however stringent inclusion criteria were adopted which resulted in only thirteen studies for review. These studies, which incorporated at least four hours of Outdoor Learning per week for a minimum of two months, indicated that Outdoor Learning can advance students in the physical, psychological, cognitive, and social dimensions.

Standing on this foundation of initial research into the emerging field of Outdoor Learning, the authors set out to undertake a systematic review of international literature on Outdoor Learning benefits to students’ academic and socio-emotional development. In order to spread the net wider than the previous systematic review [28], our minimum time requirement will be one lesson per 1–2 week timetable cycle or annually for multi-day residential programs. Programs for all school-aged (5–18 year old) students will be considered, however, they will need to include structured learning outcomes as distinct from (equally valuable) outdoor recreation and outdoor play. The learning needs to take place in a natural outdoor space on or off the school campus (but not regular Physical Education lessons or sports sessions), so classes held in community facilities like museums will be excluded. Full details of the systematic review criteria have been registered with PROSPERO [29]. We hope to paint the brush strokes which will reveal the picture of how Outdoor Learning is helping students across the world to flourish and connect learning with the natural world around them. Our aim is similar to Beames, Higgins, and Nicol [16]—to help teachers reflect on what can be taught outside and how those subjects can be taught there most effectively.

This systematic review protocol is both timely and prescient and aims to determine the characteristics of Outdoor Learning that lead to beneficial outcomes. These may include the frequency of connection with the natural world, activities that foster executive function, structuring Outdoor Learning, cognitive growth and the development of new skills. Teacher confidence and skill level [21] may also be a critical factor for learning effectiveness in outdoor settings.

### 1.3. Efficacy of Outdoor Learning

The term ‘Outdoor Learning’ has been used in the literature to describe a range of educational activities from neighbourhood nature play settings to lessons formally embedded within school curriculum [30,31,32,33].

A number of mechanisms through which contact with nature influences whole child development have been documented [34,35,36,37,38,39]. Direct contact with nature is theorised to facilitate children’s innate capabilities through the provision of opportunities to engage in experiential, cooperative teamwork [40,41,42], and imaginative and curiosity driven student-centred learning [18,43]. A comprehensive systematic review found positive associations between children spending time in nature and their socio-emotional development, however the review noted varied quality of research [44]. The longitudinal benefits of Outdoor Learning are numerous, building upon the previously mentioned physical, cognitive [45] and emotional health benefits in general [46,47]. Outdoor Learning has been proven to foster communication, reasoning, and interactional abilities [48,49], whilst also enhancing 21st century skills such as resilience [50,51], collaboration [52], conflict resolution, and self-regulation [53,54,55,56,57,58,59].

Additional benefits attributed to participation in Outdoor Learning include: building a sense of identity, analytical skills, life ownership, stress relief [60], and increasing social cohesion [61,62,63,64,65,66]. These benefits are particularly pertinent for contemporary education with beneficial impacts for mental health and wellbeing. Outdoor Learning settings allow risk-taking to be incorporated into learning experiences, which help children develop pro-social behaviours and personal executive functioning [67,68,69].

The potency of Outdoor Learning is being underpinned by a mounting number of literature reviews that highlight the evidence-based research for the developmental and wellbeing benefits on children and adolescents [28,69,70,71,72,73,74]. Notwithstanding this trend, the effect of Outdoor Learning on academic metrics remains under-researched [75]. Indeed, many outdoor educators lament one of the key factors limiting Outdoor Learning from taking a greater role in mainstream education is the paucity of evidence demonstrating its impact on academic curriculum performance [11,76,77,78,79].

## 2. Method

The Preferred Reporting Items for Systematic Reviews and Meta-Analyses (PRISMA) guidelines [80] will be used for this systematic review.

### 2.1. Searches

The international peer-reviewed literature included in this systematic review will be limited to literature published between 2000 and 2020, which has been published in English.

Academic searches will be conducted using the following databases: (1) Education Resources Information Center (ERIC) (2) Proquest, (3) PsycINFO, (4) PubMed, (5) Sage, (6) Scopus, (7) Web of Science, and (8) Wiley. Forward citation searches will be performed on all included studies. Grey literature will also be accessed using Google Scholar (first 50 hits) and OpenGrey. The title and abstract of initial search hits will be screened by one reviewer according to the inclusion criteria, with a sample being verified by a second reviewer. Potentially relevant articles will then be read independently by two reviewers.

For a detailed protocol and search strategy, please refer to our registered and published protocol under the International Prospective Register of Systematic Reviews (PROSPERO) Number CRD42020153171.

### 2.2. Search Strategy and Terms

An interdisciplinary team with a diverse range of expertise (including specialization in education, public health, psychology, and early childhood) was consulted in the development of this strategy, which should lead to an exhaustive and relevant search process. The specific search terms are listed in Table 1 under three headings, “Education Outside the Classroom”, “Learning” and “Wellbeing”, which will be combined in the search process with “AND”.

The search strategy will be adapted as appropriate for each database, with searches conducted on title and abstract. Included studies will be stored on Endnote reference management software, and initial screening decisions will be made using Rayyan software.

### 2.3. Article Screening

The first stage of screening articles’ title and abstract will delete non-English language papers, duplicate articles, and other studies clearly identified as inappropriate according to the criteria of this review. Quantitative and qualitative research methods will be considered, including randomized controlled trials (RCTs), quasi-experimental, case-control, pre-post, cross-sectional, and cohort studies.

One reviewer will screen all records for inclusion, and a second reviewer will independently screen 10% of records early in the process. Any discrepancies will be discussed in order to determine the relevance of that study, and to refine the inclusion criteria for the remainder of the screening process.

#### 2.3.1. Inclusion Criteria

The focus of this systematic review is research that investigates the impact of intentional Outdoor Learning. Included programs must report on measurable learning outcomes and be conducted in an outdoor setting (either on school grounds or in the community). The program must involve school-aged students spending regular or extended time in an outdoor environment rather than a one-off excursion, and there needs to be a connection between the characteristics of the physical setting and the learning outcomes.

#### 2.3.2. Exclusion Criteria

Studies that do not fulfil all the inclusion criteria will be removed from the review. Specific exclusion criteria include non-school aged participants, indoor off-campus environments, no identifiable learning outcomes, unstructured outdoor play, and poor research quality. Papers need to report on original research, therefore, secondary sources such as commentaries, theoretical papers, literature reviews, opinion, and editorial articles will be excluded. Case studies and conference proceedings will also be excluded from the review.

### 2.4. Article Evaluation

The second stage of the review involves two reviewers independently reading the full text of all records which remain after the screening stage, to confirm that they meet the inclusion criteria. Similarly to the screening stage, any disagreements will be resolved through dialogue between the two reviewers, or if needed, deferment to a third reviewer. 

### 2.5. Data Extraction and Analysis

Data from the final set of included studies will be categorised using a bespoke Excel spreadsheet, including: publication year, author(s), national/regional context, funding sources, research objectives, sample/population characteristics (e.g., gender, age, particular features), research method(s), location type (school campus, urban, natural), type of learning activity (academic class, outdoor education, etc.), program characteristics (e.g., duration, number of participants), measured outcomes (academic, socio-emotional), comparison environment or activity if applicable, and main findings.

#### Quality Assessment

Similarly to the data extraction process, one reviewer will apply an appropriate risk of bias assessment tool for each included study, and a second reviewer will independently assess risk of bias for 10% of studies to ensure consistency.

The quality of quantitative studies will be appraised using the CCEERC Quantitative Research Assessment Tool [81] which assesses twelve research quality factors:(1)Population(2)Randomised selection of participants(3)Sample size(4)Response and attrition rate(5)Main variables or concepts(6)Operationalization of concepts(7)Numeric tables(8)Missing data(9)Appropriateness of statistical techniques(10)Omitted variable bias(11)Analysis of main effect variables(12)Ethics approval

The quality of qualitative studies will be appraised using the JBI Checklist for Qualitative Research [82], which assesses ten questions:(1)Is there congruity between the stated philosophical perspective and the research methodology?(2)Is there congruity between the research methodology and the research question or objectives?(3)Is there congruity between the research methodology and the methods used to collect data?(4)Is there congruity between the research methodology and the representation and analysis of data?(5)Is there congruity between the research methodology and the interpretation of results?(6)Is there a statement locating the researcher culturally or theoretically?(7)Is the influence of the researcher on the research, and vice- versa, addressed?(8)Are participants, and their voices, adequately represented?(9)Is the research ethical according to current criteria or, for recent studies, and is there evidence of ethical approval by an appropriate body?(10)Do the conclusions drawn in the research report flow from the analysis or interpretation of the data?

### 2.6. Data Synthesis

It is anticipated that the included studies will be diverse in methods and results, which will preclude a quantitative meta-analytic approach. Instead, a narrative synthesis of the included papers will be offered, including development of a preliminary synthesis of findings, exploration of relationships within the data, and an assessment of the robustness of the synthesis [83].

## 3. Limitations

The stated limitation of this systematic review to articles published in English may exclude pertinent studies written in other languages. This review is focused on school aged children, and therefore will omit research data from the early childhood sector internationally, where adoption of Outdoor Learning has been in place for a number of years and has resulted in many interesting studies.

The strength of systematic reviews is their clearly defined criteria and explicitly stated search terms, however, it is possible that the selected criteria miss other pertinent key terms which relate to the topic area. It is hoped that the broad list of search terms utilized in this review will capture the vast majority of relevant research in the area of school-based Outdoor Learning.

Physical education often occurs in an outdoor environment, but has been excluded from the parameters of this review. Although the distinction could be made between natural settings and ‘hardened’ outdoor spaces (e.g., an oval) which have been modified for human use, there is no doubt that students benefit simply from being outside and undertaking physical activity [84]. However, the learning focus of the outdoor component of physical education is on specific sports and the skills they require, and outdoor settings merely provide an appropriate playing surface and area for the target sport rather than being an fundamental element of learning. For example, volleyball skills could be taught just as effectively at an indoor or outdoor court.

## 4. Discussion

The systematic review protocol set out in this paper outlines a transparent and reproducible procedure for assessing the cognitive and socio-emotional benefits of Outdoor Learning. The research protocol will also provide an audit of research quality in the area of Outdoor Learning, reducing risk of bias and selective outcome reporting in individual studies.

A significant proportion of Outdoor Learning occurs in natural settings, and there is a discrete body of literature on the benefits of ‘nature connection’ [85,86]. It was decided that the parameters of this review would include human-made outdoor settings (e.g., a zoo or cityscape) as well as natural contexts, and that the synthesis of research would be sensitive to differences in outcomes across various built and natural environments.

This review will bring together research on Outdoor Learning initiatives arising from distinct local contexts, to provide an international audit in this important field. Coalface educators, and even local policy makers, may only be aware of their particular iteration of Outdoor Learning, however this review will broaden their knowledge of various forms of Outdoor Learning, and help them to appreciate how their efforts fit into an international movement toward incorporating outdoor environments into the learning landscape.

## 5. Conclusions

The systematic review set out in this paper will produce a thorough exploration of Outdoor Learning outcomes for school children. The review will synthesize emerging evidence about the academic and socio-emotional benefits of education outside the classroom, and broaden understandings of holistic educational outcomes. It will also provide some commentary on the common themes which underly effective practice in Outdoor Learning.

The findings of this review will suggest relevant focus areas for future research and educational programs, by highlighting program and pedagogical characteristics which are most beneficial across contexts, as well as promising areas in which further research is needed. This synthesis of effective factors in Outdoor Learning will allow governments and policy makers to be better informed as they work with curriculum designers and education practitioners to utilize natural environments for maximising holistic student outcomes.

## Figures and Tables

**Table 1 ijerph-18-01199-t001:** Proposed search terms.

Education Outside the Classroom	Learning	Wellbeing
Adventure education	Academic	Communication
Adventure learning	Creativity	Cooperation
Education outdoor *	Cross curriculum	Decision making
Education outside the classroom	Educat *	Equity
Environmental education	Embodied	Health
Environmental learning	Environment *	Independence
Experiential education	Environ * stewardship	Interpersonal
Experiential learning	Experiential	Justice
Forest school	Health literacy	Life skills
Friluftsliv	Learning	Personal development
Learning outdoor *	Mastery	Pro-social
Learning outside the classroom	Physical literacy	Prosocial behave *
Nature school	Proficiency	Resilien *
Natural learning	Risk *	Psychosocial
Natural connections	Socialization Socialisation	Psycho-social
Place-based education	Sustainab *	Self confidence
Place-based learning	Trust	Self esteem
Teaching outdoor *		Soci *-emotional
Udeskole		Teamwork
		Wellbeing

* This abbreviation allows for variation in spelling or suffixes.

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
