# Peer review of "A Systematic Review Protocol to Identify the Key Benefits and Efficacy of Nature-Based Learning in Outdoor Educational Settings"

_ijerph, 2021, doi:10.3390/ijerph18031199_

Round 1

Reviewer 1 Report

I commend the research team for this concise but well formulated paper that effectively builds the case for an exciting and relevant piece of research into outdoor learning, which I have no doubt will be of significant value to the field. The justification for this particular research approach is well presented and supported by an extensive range of literature. The proposed methods for the systematic review are clear, robust and well supported. 

I was particularly impressed with the diversity, international scope, and interdisciplinary nature of the research team. I think this brings real quality to the proposed project.

I am really exited about the proposed systematic review of the benefits and efficacy of outdoor learning. It is of high value and i am sure it will be well received by the field.

The paper was very well written. I think this is the first time I have ever recommended accepting a paper with no revisions. Well done!.

This paper was unusual in that it was not based on research results, rather it was like a research proposal paper. As such it feels like the proper paper is yet to come – when they actually complete the research.

For a research proposal this was of an acceptable standard. I know the field of outdoor learning very well and the authors discussed that area well and used a very good range of literature to support that discussion. There are always gaps – it is impossible to include everything. For a paper of this type I think what was covered was appropriate.

The proposed methodology for the systematic review was robust. The interdisciplinary international team they have assembled to conduct the research is impressive. I have every confidence they will conduct a very good research project that will be very well received by the field.

Author Response

Thank you for your constructive feedback. We appreciate your positivity.

You are correct that this paper is a protocol piece stating the need for a systematic review in this area and our intention to carry it out. The following paper will describe the findings of the systematic review in full. 

Reviewer 2 Report

The protocol which aims to identify the benefits and efficacy of nature-based learning in outdoor educational settings is interesting and valuable for environmental educators as well as for health educators and other actors who are interested in this subject. If a presentation of suggested protocols is included in the scope of this journal, I have several comments and suggestions before considering it for publication:

To my opinion, the conceptualization of outdoor learning is partial and should be further expanded. I suggest using key-publications such as:

Dillon, J., Rickinson, M., Teamey, K., Morris, M., Choi, M. Y., Sanders, D., & Benefield, P. (2006). The value of outdoor learning: evidence from research in the UK and elsewhere. School science review87(320), 107.

Joyce, R. (2012). Outdoor learning: Past and present. McGraw-Hill Education (UK).

Rickinson, M., Dillon, J., Teamey, K., Morris, M., Choi, M. Y., Sanders, D., & Benefield, P. (2004). A review on outdoor learning. Field Studies Council, Shrewsbury, UK.

Tal, T. (2012). Out-of-school: Learning experiences, teaching and students’ learning. In Second international handbook of science education (pp. 1109-1122). Springer, Dordrecht.

On lines 86—87 you note: “the authors set out to undertake a systematic review of international literature on Outdoor Learning benefits to students’ academic and socio-emotional development”. This objective seems to me broader than the objective presented earlier which focus only on academic aspects of outdoor learning: “we are curious about whether academic learning in outdoor environments could also have a positive impact for curriculum-based education.” (lines 49-50). Since the benefits of outdoor learning are not limited to academic aspects, I think that the first objective should be more accurate and include other aspects which are reviewed in this protocol. To continue this point, based on the information presented in Table 1 and in the title of the protocol, I understand that two aspects will be explored in the review – learning and well-being. I suggest sticking to these aspects whenever you describe the goals of the review.

Outdoor learning occurs in diverse settings – in natural environments as well as in other out-of-class environments (e.g., museums, zoos, other man-made surroundings). Based on the title of the protocol, I understand that authors focus in their review on natural surroundings, but it is not well indicated in the protocol itself. This decision of the authors to choose natural surroundings should be better clarified as part of the conceptualization section and as part of the goal of the review. ‘Outdoor learning’ is switched into ‘nature-based learning’ (line 125). These two concepts are not synonyms. Without explaining the decision to review only outdoor learning which occur in natural surroundings, switching between phrases is not accurate and not well-understood.

The factors included in the assessment tool are diverse and relevant to the review. To my understanding, these factors focus mainly on the quality of the publications as scientific research studies (in other words - are they “good” inquiries?).  I suggest adding factors that are related to the content of outdoor learning and its contribution to academic learning and well being (as presented in the title and goals of the protocol). I think that such factors are missing in the tool. Creating such factors requires additional thinking of more qualitative categories. For example – what types of theoretical frameworks/definitions of outdoor learning are used in these studies? Do these theoretical frameworks fit the study goals and questions? Fit the methodology…

Limitations of the review – I think that it is important to add more limitations. For example, despite the effort to cover as many keywords as possible in your search, it is reasonable to assume that not all the relevant keywords could be captured (e.g., out of class learning, physical education are missing). Another limitation is related to the type of outdoor learning reviewed. Selecting studies which focus on nature-based learning/natural environments exclude other types of studies which focus on outdoor learning in non-natural settings.

Author Response

Thanks for your feedback. We have incorporated most of your suggested references, and have clarified the distinction between learning in any outdoor environment and learning in nature based settings. Your comment on limitations was also helpful, and has been included in the relevant section.

  • We have included most of the references you suggested, including recognition of the UK Forest School movement.
  • The review will focus on both academic and socio-emotional outcomes of Outdoor Learning, as stated. The lines you referred to merely describe that outdoor education has typically focused on non-curriculum outcomes, and we are also curious about academic outcomes.
  • We have included a paragraph in the discussion section explaining that much of outdoor learning is nature based, however our review will include learning in human made and natural settings. We will be sensitive to the role of ‘place’ in our synthesis of outcomes.
  • You made an interesting point about pre-defining factors by which Outdoor Learning contributes to academic and socio-emotional outcomes, however we will identify these factors inductively as we investigate the body of evidence in the review.
  • We have added text into the limitations section as per your suggestion.

Reviewer 3 Report

The submitted manuscript is apparently the exposé of a review study that has yet to be conducted. The fact that the authors write their article in future tense indicates this as well. Results of this (intended?) review study are not presented in this paper. Thus I strongly recommend to reject this manuscript.

Author Response

Thanks for your feedback.

The current paper is not a completed systematic review, as you pointed out, but rather is a protocol paper describing the need for a review and defining the parameters.

Reviewer 4 Report

The manuscript is of potential interest to the readership of this journal.

The literature is analyzed and critically appraised.

I would like more elements about your protocol and search strategy.

The results section requires far greater organisation and structuring.

I encourage you to push your analysis a bit more in each section.

Please add a discussion section.

Author Response

Thanks for your feedback.

  • The protocol and search strategy follows a similar format to a previously published protocol paper in an MDPI journal, which some of our authorship team contributed to, so we think that this level of detail is appropriate.
  • The results section also follows a similar format to a previously published protocol paper in an MDPI journal, so we think that this structure is appropriate.
  • We are not sure what you mean by ‘pushing the analysis in each section’, however acknowledge that this is the initial protocol paper rather than the completed systematic review. The aim of this paper is to describe the need for a review in this area and define the parameters of that review.
  • We have added a discussion section as suggested.

Round 2

Reviewer 2 Report

My previous comments were nicely and clearly addressed by the authors.

To my opinion the manuscript is ready for publication in its current form.

Author Response

Thanks for your feedback

Reviewer 3 Report

As the authors agree, their manuscript reports on a systematic review that has not yet been completed. The manuscript therefore appears to me almost like an internal interim report, lacking clarity as to what other researchers can learn from it. Thus, I maintain my recommendation to reject the manuscript. However, I would strongly encourage the authors to submit a paper as soon as their study is completed and then to report the final results.

Author Response

We thank the reviewer for their continued feedback. Their issue appears to be whether it is appropriate to publish a protocol to 'mark out the space' for an upcoming systematic review.

We refer them to these examples of recently published SR protocols in this journal:

Nyadanu, S.D.; Tessema, G.A.; Mullins, B.; Kumi-Boateng, B.; Bell, M.L.; Pereira, G. Ambient Air Pollution, Extreme Temperatures and Birth Outcomes: A Protocol for an Umbrella Review, Systematic Review and Meta-Analysis. Int. J. Environ. Res. Public Health 202017, 8658.

Kushemererwa, D.; Davis, J.; Moyo, N.; Gilbert, S.; Gray, R. The Association between Nursing Skill Mix and Mortality for Adult Medical and Surgical Patients: Protocol for a Systematic Review. Int. J. Environ. Res. Public Health 202017, 8604.

Mandoh, M.; Mihrshahi, S.; Cheng, H.L.; Redfern, J.; Partridge, S.R. Adolescent Participation in Research, Policies and Guidelines for Chronic Disease Prevention: A Scoping Review Protocol. Int. J. Environ. Res. Public Health 202017, 8257.